# Anaerobic Co-Digestion of Cattle Manure and Brewer's Residual Yeast: Process Stability and Methane and Hydrogen Sulfide Production

Luana Alves Akamine [1,*,†], Roberta Passini [1], João Antônio Silva Sousa [2], Aline Fernandes [2] and Maria Joselma de Moraes [1]

1 Agricultural Engineering Department, Goiás State University (UEG), Anápolis 75132-903, Brazil; rpassini@ueg.br (R.P.); mjmoraes60@gmail.com (M.J.d.M.)
2 Rural Engineering Department, São Paulo State University (UNESP), School of Agronomical Sciences, Botucatu 18610-034, Brazil; aline.nands.af@gmail.com (A.F.)
* Correspondence: luana.a.akamine@gmail.com
† This paper is a part of the master's Thesis of Luana Alves Akamine, presented at Universidade Estadual de Goiás (Brazil).

**Abstract:** Anaerobic co-digestion (AcoD) of animal waste and agro-industrial by-products has been widely studied and employed to increase biogas production potential and enhance process stability. This study evaluated the AcoD of cattle manure (CM) and brewer's residual yeast (RY) in semi-continuous biodigesters, focusing on energy potential (biogas and methane yields) and process stability. Four treatments were assessed, each with different proportions (% of volatile solids) of CM and RY: 100:0, 88:12, 78:22, and 68:32. Trials were conducted in 30-L tubular reactors at room temperature with a hydraulic retention time of 30 days. The inclusion of RY led to a gradual rise in biogas and methane production, with more significant reductions in solid content than mono-digestion of CM. The addition of RY resulted in daily $CH_4$ production increases of 18.5, 32.3, and 51.9% for treatments with 12, 22, and 32% of RY, respectively, compared to the control treatment. Therefore, AcoD demonstrated a higher potential for energy recovery. However, RY introduced elevated $H_2S$ levels in the biogas. Caution is advised when adding this co-substrate to AcoD due to potential process influence and biogas application restrictions.

**Keywords:** anaerobic co-digestion; bovine manure; brewing yeast; brewery waste; methane





## 1. Introduction

Agribusiness and its sectors are vital contributors to the Brazilian economy, although they generate vast amounts of waste. This waste holds the potential for energy production through biogas, presenting an opportunity for value addition. According to data from the International Center for Renewable Energies—Biogas, there were 885 plants in operation in Brazil in 2022. The agricultural sector accounted for 78% of the biogas plants in operation in Brazil. In contrast, the industrial and sanitation sectors contributed 12 and 10%, respectively, to the plant count. Regarding biogas volume, the sanitation sector was the primary contributor, responsible for 74% of the total volume produced. The industrial and agricultural sectors followed, contributing 16 and 10%, respectively [1]. However, the volume used represents only 3.3% (2.8 billion $Nm^3$ of biogas) [1] of the country's total potential (84.6 billion $m^3$ of biogas/year) [2].

Cattle manure (CM) has a highly biodegradable fraction [3] and a high buffering capacity, factors that optimize the anaerobic digestion (AD) process [4–6]. Due to the high-fiber diet fed to cattle, another fraction of manure is characterized by a high lignocellulosic content that is difficult to degrade [7–9]. Hence, CM may have lower yields of biogas ($m^3$ $kgVS^{-1}$) and, consequently, methane [3,5], compared to other manures such as swine and poultry. The challenges of AD in CM are often related to the low C/N ratio [5,8] and

the high level of ammonia formed during protein degradation [8,9]. However, ammonia nitrogen at the optimum concentration can be beneficial to bacteria as a source of nutrient nitrogen for microbial cell growth [10].

Previous studies have indicated that the anaerobic co-digestion (AcoD) of cattle manure with other co-substrates (e.g., agro-industrial by-products [11], agricultural waste, and energy crops [6,12]) as a carbon source results in higher methane yields, which is a direct economic benefit compared to the anaerobic mono-digestion of the manure [7,12–14]. Furthermore, AcoD utilizing multiple raw materials contributes positive interactions and synergies to these processes, including buffering capacity, balanced nutrient compositions, and an improved C/N ratio, which can enhance process stability [4,5,15–18].

Residual yeast (RY) from breweries is rich in biodegradable organic matter, offering a notable opportunity for energy generation through its conversion into biogas [19–24]. Residual yeast primarily contains carbon chains, including proteins (47.2%) [25] and carbohydrates (21.5–35.1%, dry basis) [25,26]. Its composition also encompasses macro- and micro-minerals (P, K, Na, Ca, Mg, Fe, Cu, etc.) [27], lipids (6.74%) [26], enzymes, and RNA (5.5–7.0%, dry basis) [25,27]. It is the second most abundant by-product in breweries, followed by grain waste. In the brewing process, the yeast metabolizes the fermentable sugars in the wort, alcohol, and carbon dioxide ($CO_2$). During alcoholic fermentation, the brewing yeast tends to multiply 3–6 times in the reactor. It is common practice in the brewing industry to reuse the brewery yeasts several times (4–6 times) to inoculate new fermentation tanks [28]. During beer production, for every 100 L of beer produced, 1.5–3.0 kg of RY is discarded, so large volumes of beer produced lead to the generation of significant amounts of residues [23,29,30]. Most RY is utilized in animal nutrition and feed formulation [30–32]. However, a sizable portion of this by-product remains to be disposed of, indicating a potential avenue for deriving value-added products such as biogas. In addition, RY stands out as an easily accessible material as it has a continuous supply and a low cost.

Previous studies have indicated that the AD of RY, targeted at methane production, demonstrates enhanced outcomes when co-digested with substrates such as brewery wastewater [33,34], brewery waste grains and glycerol [35], biochar [24], food waste [36], tofu wastewater [37], cardboard [38], and wastewater sludge [39]. Furthermore, evidence has suggested that the *Saccharomyces cerevisiae* found in RY can bolster the biodegradability of substrates known for their limited degradability [37,40]. Hence, RY is emerging as a valuable co-substrate for AD. The insights gained from these studies can aid the brewing industry in devising strategies to manage this waste, ensuring a positive environmental impact.

In support of the current energy transition in Brazil and its commitment to promoting economic growth supported by a clean energy matrix, this study aimed to evaluate the potential for bioenergy generation from the co-digestion of two wastes widely available in the country: brewery and livestock wastes. Furthermore, the addition of RY as a co-substrate to increase the methane yield of biodigesters operated with livestock waste (cattle manure) through AcoD has not yet been reported in the literature. Given these findings, this study aimed to evaluate the AcoD of CM and RY across varied ratios, focusing on understanding the energy potential (biogas and methane yields) and process stability.

## 2. Materials and Methods

### 2.1. Study Area

This study was conducted at the Biodigestion and Waste Management Laboratory at the Central Campus—Anápolis Headquarters of the Exact and Technological Sciences of the Goiás State University. The climate in the region is classified as Aw according to the Köppen system. This classification denotes a rainy season from October to March and a dry season from April to September. The region has an average temperature of 22.4 °C and receives an average annual rainfall of 1586 mm.

### 2.2. Inoculum and Feedstock

The inoculum was digested in a full-scale digester (covered lagoon biodigester) treating diluted dairy cattle manure (without the solid fraction) on a semi-continuous system with stabilized biogas production and a methane content equal to $60 \pm 2\%$ in its composition. The feedstocks were dairy cattle manure (CM) and brewer's residual yeast (RY), which were collected from a dairy farm and a brewery, respectively. Both are located near the city of Anápolis in Goiás State, Brazil. The CM was collected weekly from the milking parlor and the area around the trough; both environments had a concrete floor.

### 2.3. Semi-Continuous Biodigester Description

On a laboratory scale, the semi-continuous biodigesters were made up of hermetically sealed polyvinyl chloride (PVC) pipes with two distinct parts: the container with the fermenting material (fermentation chamber) and the gasometer (Figure 1). Semi-continuous digesters have an entrance for the load and an exit for the digestate, plus a hose to conduct the biogas to the gasometer. The gasometer consisted of two PVC tubes; one external pipe with a 25 cm diameter was filled with water, and a second pipe with a 20 cm diameter was submerged in water to allow displacement by the gas produced in the fermentation chamber. A graduated ruler was fixed to the outside of the gasometer to measure the displacement of the tube. The hydraulic retention time used was 30 days, so since the useful volume of the reactors was 30 L, the daily loads (DL) were 1.0 L. The semi-continuous digesters were kept at room temperature, with an average of 25.2 °C, a maximum of 30.9 °C, and a minimum of 19.6 °C.

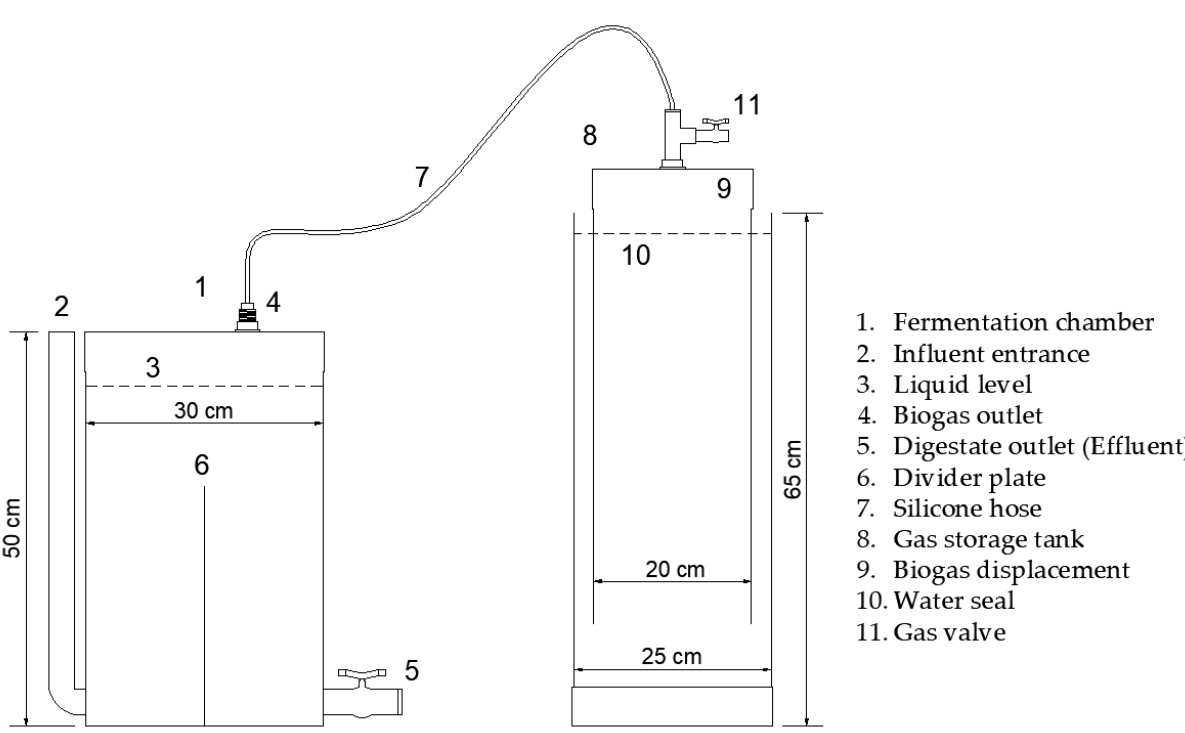

**Figure 1.** Schematic cross-sectional design of the semi-continuous biodigester.

### 2.4. Treatment Descriptions

The experiment was conducted using a completely randomized design, consisting of four treatments with four replications each, resulting in a total of 16 semi-continuous biodigesters. The treatments were based on varying proportions of dairy CM and brewery RY according to their volatile solids (VS) content. The proportions were as follows (CM/RY): 100:0, 88:12, 78:22, and 68:32.

The reactors were operating for 50 days. Initially, the biodigesters were filled using a 50/50 (*v/v*) ratio of inoculum and sieved fresh dairy CM ($\leq 3$ mm). The inoculum

adaptation took place between days 1–5, and no daily loads were carried out during this period. Between days 6–15, daily loads (1 L) were carried out only with diluted and sieved CM. Initiating the loads with the inclusion of RY was contingent upon achieving 60% methane in the biogas composition. Subsequently, over the next 30 days, RY was added in the studied proportions. The acclimatization of the anaerobic microorganisms to the substrate was expedited due to the inoculum's origin; it was sourced from the digestion of diluted and sieved manure, which matched the conditions and characteristics of the DLs employed in the experiment.

The CM was diluted with water at a 1:4 ratio to prepare the DLs, aiming for a total solids (TS) concentration of 4%. The diluted CM was then sieved through a 3-mm mesh to remove fibrous materials ($\geq$3 mm). By the end of the DL preparation, the content was adjusted to approximately 2% TS (OLR = 0.36–0.47 g VS L $d^{-1}$). The characteristics of the materials used in the co-digestion are listed in Table 1.

**Table 1.** Characterization of the substrates used in the semi-continuous biodigester experiment.

| Parameters | | Fresh Cattle Manure | CM * | RY |
|---|---|---|---|---|
| pH | pH unit | 6.10 ± 0.20 | 6.88 ± 0.27 | 4.30 ± 0.21 |
| Total solids | % (NM) | 15.10 ± 0.79 | 1.51 ± 0.13 | 17.63 ± 0.47 |
| Volatile solids | % of the TS | 84.37 ± 2.64 | 72.95 ± 2.96 | 97.80 ± 0.80 |
| Ash | % of the TS | 15.70 ± 3.33 | 26.82 ± 2.79 | 2.20 ± 0.47 |
| COD | g $O^2$ $L^{-1}$ | 58.42 ± 8.79 | 12.09 ± 0.8 | 268.96 ± 35.10 |
| TOC | % of the TS | 46.87 ± 1.46 | 40.53 ± 1.64 | 54.33 ± 0.45 |
| TKN | % of the TS | 3.00 ± 0.51 | 1.61 ± 0.14 | 7.11 ± 0,12 |
| Phosphorus | % of the TS | - | 2.45 ± 0.09 | 0.95 ± 0.01 |
| Potassium | % of the TS | - | 1.51 ± 0.02 | - |
| C/N ratio | dimensionless | 15.62 ± 0.40 | 25.21 ± 1,02 | 7.64 ± 0.18 |
| NDF | % of the TS | - | - | 10.7 ± 0.20 |
| ADF | % of the TS | - | - | 6.8 ± 0.16 |
| Protein | % of the TS | - | 10.06 ± 0.10 | 44.44 ± 0.09 |

* Diluted cattle manure without the solid fraction. NM: natural matter; TS: total solids; ; COD: chemical oxygen demand; TOC: total organic carbon; TKN: total Kjeldahl nitrogen; C/N: carbon/nitrogen ratio; NDF: neutral detergent fiber; ADF: acid detergent fiber; -: not determined; (mean ± standard deviation, *n* = 4).

Covered lagoon biodigesters are commonly used to treat livestock waste in Brazil (especially cattle and pig waste). However, this biodigester model has some limitations, such as limited volumetric organic load (0.3–0.5 kg SV $m_{reactor}^{-3}$ $d^{-1}$) and high hydraulic retention time (30–45 days). The use of preliminary solid separation techniques increases methane production capacity and can reduce the size of the treatment plant [41]. Generally, a solid fraction of manure is submitted to the composting process.

The composition of the daily feedings for the treatments was as follows: RY 0%: 96.47 g CM + 903.50 g water; RY 12%: 91.70 g CM + 8.51 g RY + 899.80 g water; RY 22%: 86.80 g CM + 17.02 g RY + 896.20 g water; and RY 32%: 82.00 g CM + 25.52 g RY + 892.50 g water. During the preparation of the DLs, the pH of the RY was adjusted using sodium bicarbonate.

*2.5. Analytical Methods*

The influents and effluents of the treatments, as well as the CM and RY, were characterized by physico-chemical analysis. The TS, VS, and ash content were quantified using gravimetric methods, which involved drying and igniting the sample [42]. The pH was measured using a pH meter (KASVI, model K39-0014P). Partial alkalinity (PA) and intermediate alkalinity (IA) were determined by the titration method [43]. The total alkalinity was calculated from the sum of PA and IA. Total Kjeldahl Nitrogen (TKN) was determined after sulfuric acid digestion, followed by distillation using a Kjeldahl distiller. The distillate was then titrated with $H_2SO_4$ [42]. Total organic carbon (TOC) was estimated by dividing the percentage of VS by 1.8 [44]. The C/N ratio was derived from the ratio between TOC and TKN.

Samples were digested in a nitric-perchloric solution with an external heat source to determine phosphorus (P) and potassium (K) levels, followed by dilution and filtration. Concentrations of P were determined using absorbance readings from a digital spectrophotometer, while K was quantified via flame photometry [45].

### 2.6. Biogas Monitoring

The volume of biogas produced was determined by measuring the vertical displacement of the gasometers and then multiplied by the internal cross-sectional area of the gasometers ($0.02956$ m$^2$). Temperature, biogas volumes, and environmental conditions were monitored throughout the experimental period. The correction of biogas volume for the conditions at 1 atm and 20 °C was carried out by means of Equation (1), resulting from the combination of Boyle and Gay-Lussac laws.

$$\frac{(V_0 \times P_0)}{T_0} = \frac{(V_1 \times P_1)}{T_1} \tag{1}$$

where:

$V_0$—Corrected volume of the biogas, m$^3$;

$P_0$—Corrected pressure of the biogas, 10,322.72 mmH$_2$O;

$T_0$—Corrected temperature of the biogas, 293.15 Kelvin (K);

$V_1$—Volume of the gas in the gasometer;

$P_1$—Biogas pressure at the time of reading, in mm H$_2$O;

$T_1$—Biogas temperature at the time of reading, in K.

The biogas composition, including CH$_4$, CO$_2$, and H$_2$S, was analyzed weekly using a portable gas analyzer (Gasboard-3200L) (Figure 2). The analyzer uses dual-beam (nondispersive infrared) detectors for CH$_4$ and CO$_2$ analysis and industrial electrochemical cells for H$_2$S analysis.

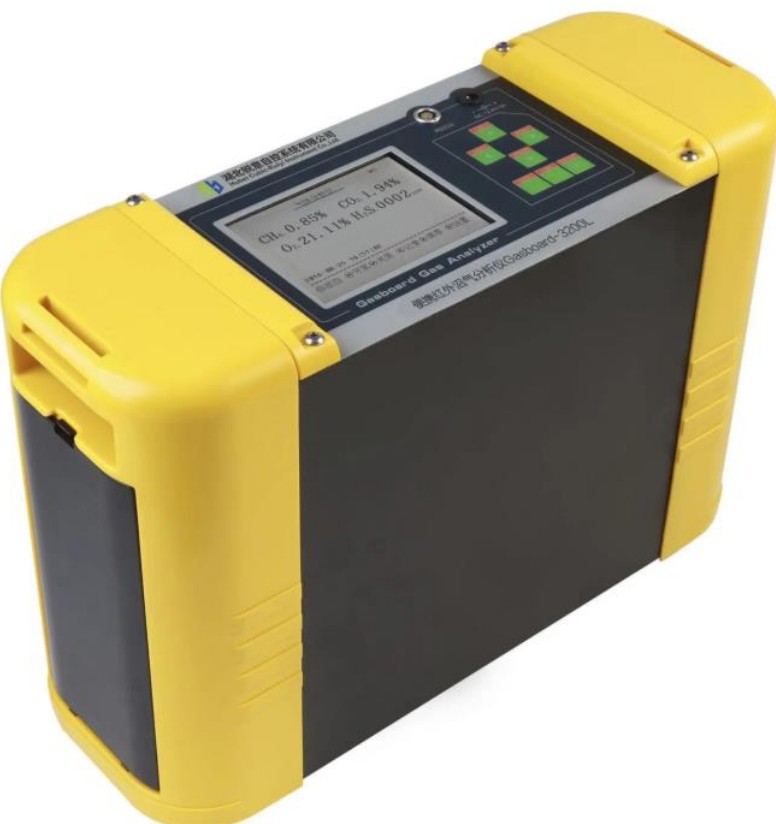

**Figure 2.** The portable gas analyzer employed in this study: a Gasboard-3200L.

Specific biogas (SBP) and methane production (SMP) were determined using the biogas production data (L) and TS and VS added (kg) to the biodigesters during anaerobic digestion. The values were presented in L of biogas and methane per kg of TS and VS added (L kgTS$_{added}$$^{-1}$ and L kgVS$_{added}$$^{-1}$, respectively).

*2.7. Statistical Analysis*

The data was subjected to an analysis of variance using the F-test, and when significant, the means were compared using Tukey's test at a 5% significance level. The homogeneity of the variances and the normality of the residuals were verified as assumptions. The data were subjected to regression analysis. Multivariate analyses employed hierarchical grouping analysis (cluster analysis) and principal component analysis (PCA). Once obtained, the degree of influence was verified and interpreted through the correlation between each characteristic. Using the scores of the principal components, a two-dimensional scatter plot was generated to visualize the dispersion of each treatment [46]. Statistical analyses were conducted in the R program, utilizing the 'MultivariateAnalysis' functions [47].

**3. Results and Discussion**

The characteristics of both the influent and effluent from the anaerobic co-digestion of CM and the brewer's RY are presented in Table 2.

**Table 2.** Characterization of the influent and effluent from the anaerobic co-digestion of cattle manure and brewer's residual yeast in semi-continuous digesters.

| Parameters | | RY | | | |
|---|---|---|---|---|---|
| | | 0% | 12% | 22% | 32% |
| pH | Influ. | 6.88 ± 0.27 | 6.97 ± 0.25 | 7.07 ± 0.21 | 7.19 ± 0.21 |
| | Efflu. | 7.16 ± 0.09 | 7.19 ± 0.09 | 7.33 ± 0.30 | 7.39 ± 0.29 |
| TS (g L$^{-1}$) | Influ. | 15.06 ± 1.34 | 17.43 ± 0.77 | 18.09 ± 1.71 | 19.02 ± 1.09 |
| | Efflu. | 7.72 ± 0.63 | 6.43 ± 0.01 | 6.55 ± 0.31 | 6.47 ± 0.58 |
| VS (g L$^{-1}$) | Influ. | 10.96 ± 0.81 | 12.41 ± 0.57 | 13.22 ± 0.86 | 14.24 ± 0.65 |
| | Efflu. | 4.32 ± 0.40 | 3.55 ± 0.07 | 3.48 ± 0.21 | 3.29 ± 0.36 |
| VS/TS | Efflu. | 0.56 ± 0.01 | 0.55 ± 0.01 | 0.53 ± 0.02 | 0.51 ± 0.01 |
| TOC (g L$^{-1}$) | Influ. | 6.09 ± 0.45 | 6.60 ± 0.43 | 7.11 ± 0.41 | 7.62 ± 0.38 |
| | Efflu. | 1.34 ± 0.14 | 1.09 ± 0.04 | 1.03 ± 0.08 | 0.93 ± 0.12 |
| NKT(g L$^{-1}$) | Influ. | 0.24 ± 0.02 | 0.34 ± 0.02 | 0.43 ± 0.02 | 0.53 ± 0.02 |
| | Efflu. | 0.10 ± 0.01 | 0.71 ± 0.01 | 0.71 ± 0.03 | 0.70 ± 0.06 |
| C/N ratio | Influ. | 25.17 ± 1.02 | 19.59 ± 0.55 | 16.47 ± 0.42 | 14.48 ± 0.35 |
| | Efflu. | 13.27 ± 0.34 | 1.53 ± 0.06 | 1.45 ± 0.08 | 1.32 ± 0.07 |

TS: total solids; VS: volatile solids; TOC: total organic carbon; NKT: nitrogen Kjeldahl total.

The recommendation for AcoD with CM is primarily due to its nitrogen content, potentially offering a more favorable C/N ratio for the AD process [4,12]. In this study, the CM underwent sieving and, therefore, had a lower N content (1.61%) than the fresh manure (3.00%) (Table 1). Conversely, RY is rich in N (7.11%), and even though it reduced the C/N ratio, it increased the substrate's carbon content.

An increasing linear trend concerning RY additions was observed for the parameters, biogas and methane yields, and solids reduction (Figure 3). Reductions in solid constituents (TS and VS) confirmed the biogas and methane yields (L kg$^{-1}$ per kg of TS$_{added}$ and VS$_{added}$), showing that the addition of RY was beneficial according to the prediction models.

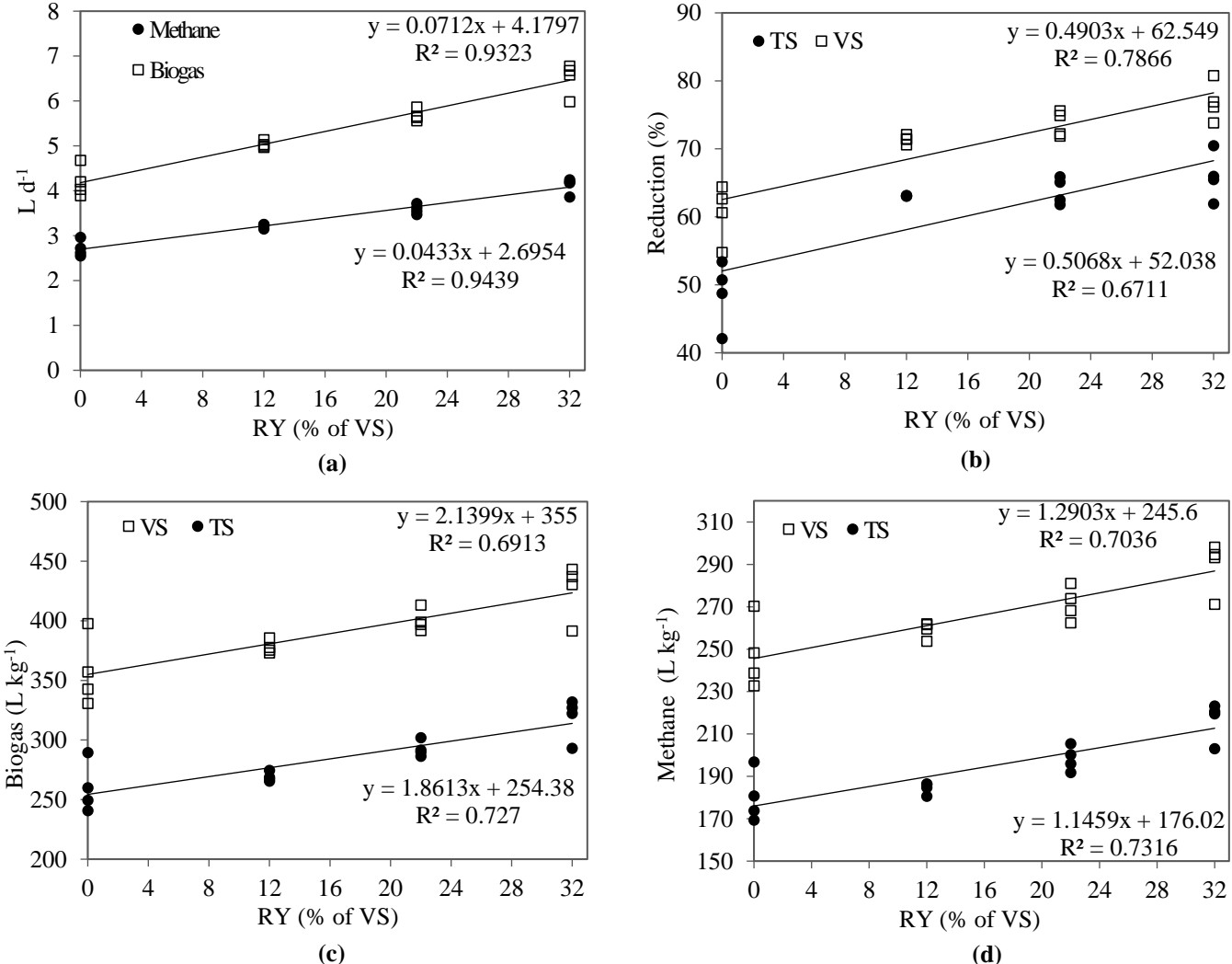

**Figure 3.** Daily biogas and methane yield (**a**), solids reduction (**b**), specific biogas yield (**c**), and specific methane yield (L per kg of $TS_{added}$ and $VS_{added}$) (**d**) obtained from the treatments.

The main reason for the increase in methane yields (Figure 3a,d) is the high biodegradability of the RY [23,24]. The increased RY in the substrate led to a lower SV/TS ratio (Table 2) and an increase in SV reduction, reflecting the degradation of the available organic matter. As Sosa-Hernandez et al. [21] highlighted, RY is replete with proteins, amino acids, and carbohydrates, which can be hydrolyzed into soluble chemical oxygen demand (CODs). The high solubilization rate of these compounds favors the use of the substrate by microorganisms and demonstrates that yeast is a suitable substrate for biogas production [35].

The increases in solids reductions were 29.4, 30.9, and 35.3% for TS and 17.8, 21.5, and 26.9% for VS for the 12, 22, and 32% RY inclusions, respectively, compared to the 0% RY (Table 3). The reduction in solids, especially VS, is intrinsically linked to the bacteria's utilization of organic matter to produce biogas. As anticipated, the reductions in solids paralleled the yields observed (i.e., the specific biogas and methane production increased with the increased proportion of RY in the substrate).

No studies were found in the literature evaluating the AcoD of CM and RY. Tewelde et al. [48] investigated the AcoD of CM combined with brewery waste in a batch process (8% TS). They deduced that a 70:30 ratio (CM/brewery waste) led to the highest methane yield (0.287 $m^3$ $kgVS_{added}^{-1}$) and the maximum methane content in the biogas (69%).

**Table 3.** Daily and specific biogas and methane production, solids reduction, methane and hydrogen sulfide content obtained from the treatments.

| Parameters | | RY | | | | $p$ | CV |
|---|---|---|---|---|---|---|---|
| | | 0% | 12% | 22% | 32% | | |
| Biogas | L d$^{-1}$ | 4.2 d | 5.0 c | 5.7 b | 6.5 a | <0.001 | 4.78 |
| | L kgTS$_{added}$$^{-1}$ | 260.0 c | 269.1 bc | 292.5 ab | 318.7 a | 0.0011 | 4.99 |
| | L kgVS$_{added}$$^{-1}$ | 357.2 c | 378.1 bc | 400.3 ab | 425.7 a | 0.0041 | 4.96 |
| Methane | L d$^{-1}$ | 2.71 d | 3.22 c | 3.59 b | 4.12 a | <0.001 | 4.10 |
| | L kgTS$_{added}$$^{-1}$ | 180.2 b | 184.5 b | 198.4 ab | 216.6 a | <0.001 | 4.28 |
| | L kgVS$_{added}$$^{-1}$ | 247.5 b | 259.2 b | 271.4 ab | 289.3 a | 0.0033 | 4.26 |
| TS$_{red.}$ | % | 48.7 b | 63.1 a | 63.8 a | 66.0 a | <0.001 | 5.69 |
| VS$_{red.}$ | % | 60.6 b | 71.4 a | 73.6 a | 76.9 a | <0.001 | 4.36 |
| Methane | % | 64.3 a | 64.3 a | 63.4 a | 63.2 a | 0.0667 | 1.03 |
| Hydrogen sulfide | ppm | 24.8 b | 44.1 ab | 55.3 a | 59.0 a | 0.0026 | 20.58 |

TS$_{red.}$: total solids reduction; VS$_{red.}$: volatile solids reduction; CV: coefficient of variation; Means values followed by different letters in a row significantly differ by Tukey's test ($p < 0.05$).

In an earlier study by Zupančič et al. [34], the addition of 0.7% (1.24 gVS L$^{-1}$, UASB reactor) of RY in AcoD with wastewater led to a 50% surge in biogas production. These researchers noted that while a concentration of 1.1% (1.95 gVS L$^{-1}$) did not produce adverse effects, concentrations of 1.6 and 2.3% (2.83 and 4.07 gVS L$^{-1}$) instigated process instability. A concentration of 2.8% (4.96 gVS L$^{-1}$) caused the system to collapse due to excessive solid loading and minimal degradation. The RY concentrations in the substrate assessed in this study mirrored those examined by the authors mentioned above. Yield variations can be attributed to the differential co-substrates employed (CM versus wastewater) and reactor type disparities (plug flow versus UASB). The RY concentrations of 12, 22, and 32% in the substrate (1.47, 2.93, and 4.40 gVS L$^{-1}$, respectively) resulted in daily biogas production augmentations of 19.8, 35.2, and 54.8%. Biogas and methane production discrepancies can also occur due to RY composition, which may fluctuate based on the type of beer source, material collection method, and additives such as hops [21].

Syaichurrozi et al. [37] reported that the presence of *Saccharomyces cerevisiae* in RY facilitates polysaccharide degradation under anaerobic conditions. Islas-Espinoza et al. [40] observed that including *Saccharomyces cerevisiae* was beneficial, enhancing cellulolytic activity and expediting methane production in fruit and vegetable waste AD. Akyol et al. [49] reported that fungal bioaugmentation can amplify the digestion efficiency of substrates abundant in lignocellulose. In our study, RY in the AcoD system may have increased the biodegradability of CM given that the manure has a higher fiber content. Cattle manure typically encompasses 7.39% of rapidly degraded carbohydrates and 49.01% of slowly degraded carbohydrates [3]. However, the latter's degradability may show more effectiveness or speed in AcoD in the presence of co-substrates [7].

The addition of RY did not increase ($p > 0.05$) the methane content in the biogas. Nevertheless, it increased the volume of biogas generated. This, in turn, facilitated a compensatory effect, culminating in a larger ($p < 0.05$) methane volume during AcoD (Table 3). In terms of daily methane production, the addition of RY led to increases of 18.5, 32.3, and 51.9% for inclusions of 12, 22, and 32%, respectively, compared to the control treatment (Figure 4).

An increase in hydrogen sulfide (H$_2$S) levels in the biogas was observed as the RY concentration in the substrate augmented; H$_2$S originates during the breakdown of sulfur-rich proteins [50]. According to the literature [26], RY is rich in protein (44.44% relative to dry matter; Table 1). In short, RY has sulfur amino acids [31,51], generating sulfides (S$^{2-}$, HS$^-$, and H$_2$S) in solution and H$_2$S in the biogas in the anaerobic process [52]. The reduction of sulfate to sulfide can inhibit the anaerobic process due to competition between sulfate-reducing bacteria and methanogenic archaea, as well as the toxicity of this compound,

an important inhibitor of methanogenesis, leading to reduced biogas production with comparatively lower methane levels [52–54]. Therefore, the amount of RY in the AcoD process should be meticulously scrutinized and regulated. Otherwise, an overabundance of these compounds might instigate the cessation of the AD process.

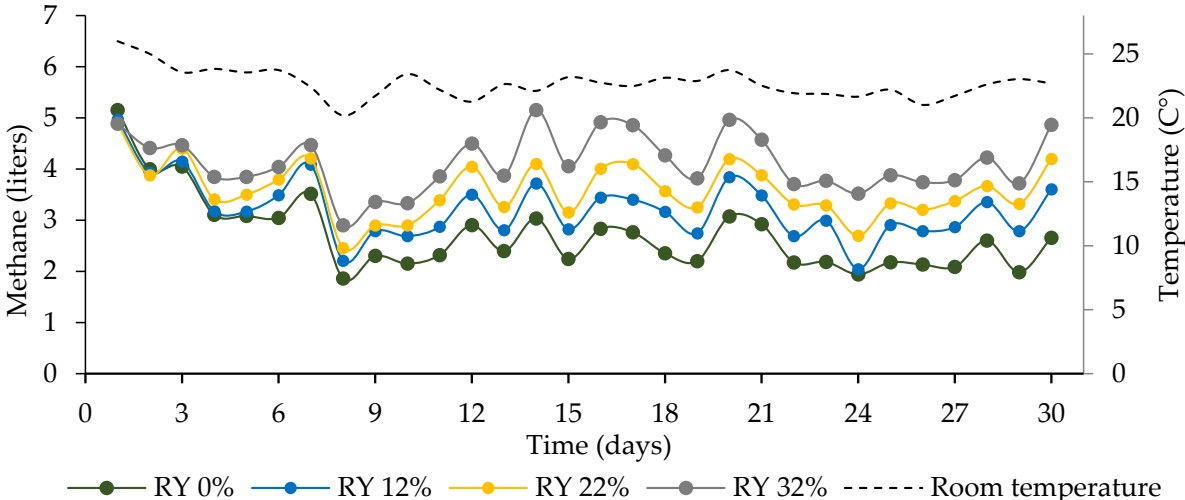

**Figure 4.** Daily methane yield obtained from the anaerobic co-digestion of cattle manure and the brewer's residual yeast in semi-continuous digesters.

The $H_2S$ is considered a potentially corrosive component and must be removed from biogas in order to preserve the safety and integrity of the equipment. In general, low values positively affect the useful life of all plants [55]. Therefore, there are minimum quality requirements for each type of biogas and biomethane application (thermal, mechanical, and electrical energy) and end-use (heating boilers, internal combustion engines, generators, injection into the biogas or natural gas grid, vehicle fuel, fuel cells, etc.). In addition, for each use, there are different technologies and purification levels [56]. According to the National Agency for Petroleum, Natural Gas, and Biofuels [57], for a gas to be considered biomethane, it must have a minimum $CH_4$ fraction of 90%, an $H_2S$ concentration limit of 10 mg m$^{-3}$, and a $CO_2$ fraction of 3%. Among other biogas applications, the recommended $H_2S$ limits are <250 ppm for boilers (heating), 545–1742 ppm for internal combustion engines, and 2–15 mg m$^{-3}$ for upgrading biogas to natural gas [58].

The total alkalinity (2693.3–3566.2 mg L$^{-1}$) (Figure 5) and pH (7.16–7.33) (Table 2) values found in the effluent for all treatments indicate a good buffering capacity for the substrates. Indirectly, these values also indicate the balance of the process (i.e., the kinetics of acid production and consumption are balanced) [5]. The recorded total alkalinity figures can be traced back to the sodium bicarbonate addition to the RY during the daily load preparations and the acid-neutralizing prowess inherent to CM [4–6]. Evidence has indicated cattle manure's pivotal role in upholding the buffering capacity in AcoD with diverse materials, including food waste [59], agricultural waste [11], energy crops [12], and sorghum [6].

Total alkalinity is the sum of partial alkalinity (PA) and intermediate alkalinity (IA). Here, PA represents alkalinity attributable to bicarbonate, while IA is due to volatile organic acids. In scenarios where PA is absent and IA predominates, the reactor is susceptible to pH oscillations, which can potentially curtail biogas generation. An optimal IA/PA ratio that augurs well for AD performance lies between 0.3 and 0.4 [60]. Therefore, the values discerned for the treatments' effluents were within this ideal range (0.32, 0.32, 0.30, and 0.27 for 0, 12, 22, and 32% RY, respectively).

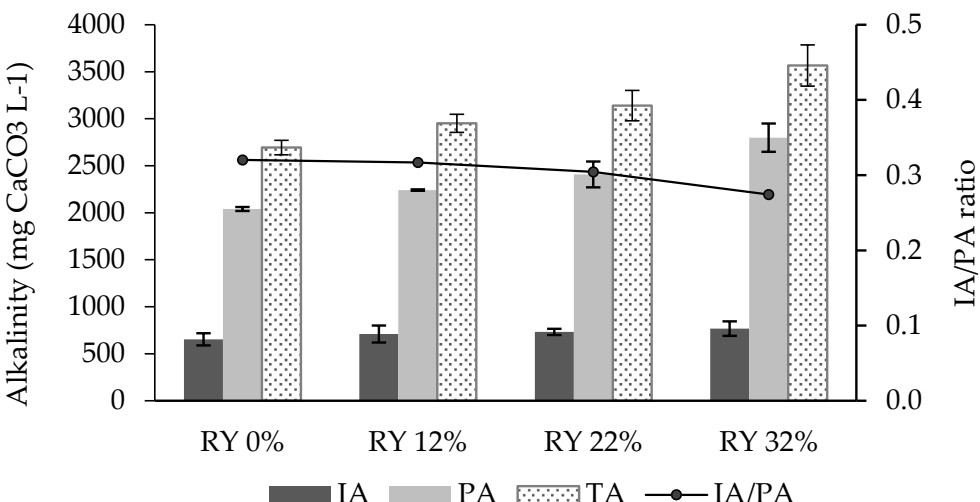

**Figure 5.** Partial alkalinity (PA), intermediate alkalinity (IA), IA/PA ratio, and total alkalinity obtained from the treatments' effluents.

*Multivariate Analysis: Clusters and Principal Components*

The multivariate analysis offered an overarching perspective on the behavior of the variables and treatments in the plan and their associations. The first two principal components (PC) of the analysis accounted for 98.53% of the total data variability (Table S1; Supplementary Materials). The biplot graph for these two PCs is depicted in Figure 6.

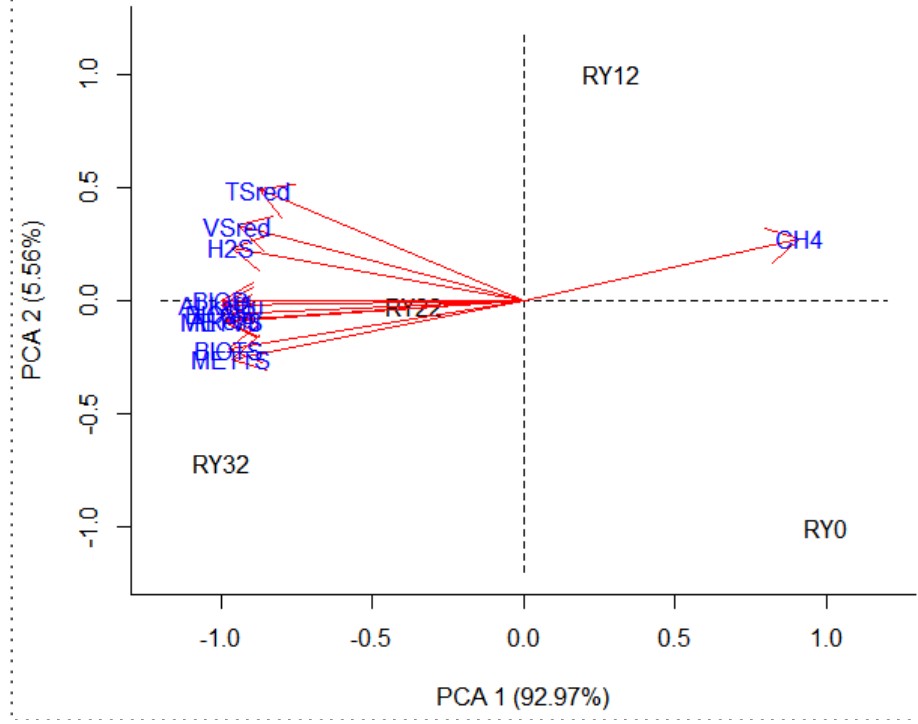

**Figure 6.** A biplot graph of the principal component analysis for the anaerobic co-digestion test. RY: residual yeast; BIOD: daily biogas production; BIOTS: specific biogas yield (L kgTS$^{-1}$); BIOVS: specific biogas yield (L kgVS$^{-1}$); METTS: specific methane yield (L kgTS$^{-1}$); METVS: specific methane yield (L kgVS$^{-1}$); TSred.: total solids reduction; VSred.: volatile solids reduction; CH$_4$: methane content in the biogas; H$_2$S: hydrogen sulfide content in the biogas; ALKinflu: alkalinity influent; ALKefflu: alkalinity effluent.

The PC1 explained 92.97% of the data variability, and it negatively correlated with the variables BIOD, BIOTS, BIOVS, METTS, METVS, Tsred, Vsred, $H_2S$, ALKinflu, and ALKefflu and positively correlated with the variable $CH_4$. PC2 only explained 5.56% of the data variability and was negatively correlated with the $CH_4$ variable. Treatments with 22 and 32% RY (lower left quadrant of the graph) clustered closer to variables associated with energy utilization, including specific biogas and methane production and solid removal efficiency (total and volatile solids).

Indeed, the treatments with 22 and 32% RY achieved the highest biogas and methane production (Table 3). It is plausible that these treatments secured balanced nutrient compositions, leading to environments synergistically optimized for the growth and performance of methanogenic microorganisms [17,61].

The quadrants, derived from the intersection of the PC1 and PC2 axes, facilitate the interpretation of the behavior of treatments concerning the analyzed variables. Hence, one can observe that the control and 12% RY treatments had the most favorable biogas compositions and reduced $H_2S$ levels. The addition of 22 and 32% of RY exhibited a slight drop in $CH_4$ content in biogas (around 1.0%) throughout the evaluation (Figure S1; Supplementary Materials). Therefore, no antagonistic effects were detected for the RY proportions examined. Nonetheless, it is conceivable that larger proportions of RY in the substrate negatively impact AcoD, resulting in the generation of inhibitory compounds such as $H_2S$ and decreasing methane levels in the biogas [53].

## 4. Conclusions

Anaerobic co-digestion conditions (22% and 32% residual yeast) were considerably more attractive in terms of energy recovery potential than mono-digestion. The additions of 12, 22, and 32% residual yeast promoted daily increases in $CH_4$ production of 18.5, 32.3, and 51.9%, respectively, compared to the control treatment.

The anaerobic co-digestion process remained stable for all proportions studied, and the addition of the brewer's residual yeast interfered with the composition of the biogas, leading to a higher $H_2S$ content. The absence of significant antagonistic effects suggests the potential for evaluating higher yeast ratios in the substrate composition remains untapped. Consequently, future research should consider these increased ratios to gain deeper insights into the yeast's impact on process stability, methane production, and overall biogas quality. Considering the abundant availability of this waste in Brazil, the findings of this study provide a compelling basis for subsequent investigations to perform a comprehensive technical and economic analysis of the co-digestion of these substrates at the studied ratios on an industrial scale. Such studies could significantly contribute to the ongoing energy transition and the valorization of bioenergy production potential.

**Supplementary Materials:** The following supporting information can be downloaded at: https://www.mdpi.com/article/10.3390/fermentation9120993/s1, Material published online alongside the manuscript, Table S1: Correlation between principal components and variables in the anaerobic co-digestion assay; Figure S1: Methane ($CH_4$, in %) content in the biogas throughout the evaluation span.

**Author Contributions:** Conceptualization, L.A.A. and R.P.; methodology, L.A.A., R.P. and M.J.d.M.; software, L.A.A. and A.F.; formal analysis, L.A.A. and J.A.S.S.; investigation, L.A.A. and J.A.S.S.; data curation, L.A.A., R.P. and A.F.; writing—original draft preparation, L.A.A., R.P., J.A.S.S., A.F. and M.J.d.M.; writing—review and editing, L.A.A., A.F. and M.J.d.M.; supervision, R.P. All authors have read and agreed to the published version of the manuscript.

**Funding:** This study received funding from "Financial Resource from Call No. 21/2022; Term of Commitment No. 000036040537; Process SEI No. 202200020020855".

**Institutional Review Board Statement:** Not applicable.

**Informed Consent Statement:** Not applicable.

**Data Availability Statement:** Data are contained within the article and Supplementary Materials.

**Acknowledgments:** This study was financed in part by the Coordenação de Aperfeiçoamento de Pessoal de Nível Superior - Brasil (CAPES) - Finance Code 001. We would also like to thank Atlas Assessoria Linguística for language editing.

**Conflicts of Interest:** The authors declare no conflict of interest.

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
