# Peer review of "Anaerobic Co-Digestion of Cattle Manure and Brewer’s Residual Yeast: Process Stability and Methane and Hydrogen Sulfide Production"

_fermentation, doi:10.3390/fermentation9120993_

Round 1
Reviewer 1 Report
Comments and Suggestions for Authors
This manuscript investigated the co-digestion of cattle manure and brewer’s residual yeast to produce more renewable energy from organic waste. Even though the main topic of this study is not novel, the contents and experimental results of this study (H2S, analysis method) are worthy of being accepted as a research article in Fermentation. However, the authors need to revise the manuscript to improve the quality of this paper according to the following comments.
- (L 38-47) There is unnecessary information (lignocellulosic content of cattle manure). It is better to remove or revise it.
- Organic loading rate (OLR) should be mentioned in the continuous operation tests. Please add OLR for the reactors in the manuscript. Additionally, the analysis results in Table 2 should be checked (especially, NKT).
- Basically, the reactors should be operated until reaching steady-state level by providing enough time for inoculum adaptation. However, the authors did not mention this in the manuscript. The operating time of semi-continuous reactors (30 d) seems to be short considering HRT. The author should show how and when the experimental data were obtained.
- Some graphs should be revised to show meaningful results. I don’t know why biogas production data are shown in the Fig 2. Is there any specific reason for showing biogas production results? I think the CH4 production data are enough to show the enhancement. Relatively, CH4 content in biogas is less important than other results such as daily CH4 production rate or yield. Therefore, Fig. 4(b) needs to be removed or revised using other data.
- (L 232-234) Obviously, CH4 production was enhanced as the portion of RY in feedstock increased. However, I can’t entirely agree that this result implies superior efficiency and synergism of co-digestion. The main reason may be just adding an easily biodegradable substrate, RT. The authors should suggest and show support for the author’s opinion.
- Conclusions should be revised. The authors' important findings, research limitations, ... should be highlighted.
Comments on the Quality of English LanguageEnglish and grammatical errors should be corrected to make the manuscript more precise and readable.
- Some sentences should be revised to help the readers understand (L 198-200, L 230-231, L 245-246, …). Is the word “affluent” in Table 2, a type error? In general, influent is used. Check line 52, “ma-terials”
Author Response
Author's Reply to the Review Report (Reviewer 1)
Dear reviewer,
All the changes made to the manuscript are highlighted in yellow throughout the text. As requested by the reviewer, the manuscript was proofread in English, and the editorial certificate is available for access. Below are the reviewer’s comments with the respective changes and comments from the authors.
Introduction:
L 40
Text removed, “(lignocellulosic content of cattle manure)”.
58-63; L 67-69; L 79-84
Two relevant references [23,24], requested by reviewer 2 (L 54), were inserted in the introduction. Both papers refer to studies on the anaerobic digestion of residual yeast.
The aim and innovation of the manuscript was revised (last paragraph of the introduction).
Materials and methods:
L 125-134 (Operating time of semi-continuous reactors)
This important point was unclear in our original article, so we inserted the requested information.
L 138
The organic loading rate (OLR) was added to the text.
Results and discussion:
Table 2. L 210
The treatments with added RY had higher amounts of N in their initial compositions and the highest organic material removal efficiencies (Table 3), which implies a higher concentration of mineral material in the effluent.
The term “affluent” was replaced by “influential.” The term was replaced throughout the text.
Figures 2 and 4
Figures 2e, 4a, and 4b (original manuscript) were removed due to data repetition.
A graph of daily methane production was added (Figure 5). Total alkalinity was presented with partial and intermediate alkalinity (Figure 4).
We chose to present the solids reduction and biogas and methane yield data in graphs, accompanied by linear regression, as this is a common analysis in trials evaluating increasing proportions.
L 228-231 (CH4 production)
The text was revised.
Conclusions:
L 364-372
The conclusions were rewritten. We inserted the main results and a new paragraph with suggestions for future studies.
We would like to thank the reviewer for evaluating our manuscript. We attempted to address all the reviewer’s concerns adequately and believe that our article improved considerably. We look forward to your final response and willing to answer any further questions.
Reviewer 2 Report
Comments and Suggestions for Authors
The article “Anaerobic co-digestion of cattle manure and brewer’s residual yeast: process stability and methane and hydrogen sulfide production” reports a laboratory investigation aimed at co-digesting locally available substrates in Brazil. The general topic of the article is in the scope of Fermentation; the article is well structured and can be of interest to a wide scientific audience. However, its novelty appears to be limited. The English language is fairly good. In my opinion, a minor revision is required before final publication, which should cover the following points:
1. Introduction: the brewery industry and the generated by-products should be better introduced, compared to what is already stated in lines 55-65. Consider 10.15255/CABEQ.2015.2237 and 10.1016/j.jece.2019.103184 as pertinent references to describe the general framework.
2. Introduction: the novelty of the present study should be better clarified at the end of the section.
3. Materials and methods: I do not understand why only the liquid fraction of cow manure was digested. How are you supposed to handle the solid fraction if you think about system upscaling? This is a critical point to be clarified.
4. Figure 1: I would say “Digestate” rather than “Biofertilizer” in bullet point 5 (and in the main text). Also, it would be nice to report a photograph of the equipment as Fig. 1b.
5. Table 1: I believe you mean “TKN”, not “NKT”.
6. Table 2: I would report also the VS/TS ratio in the Table, as it is indicative of the process efficiency. Also, the C/N ratio of the effluent is not reported.
7. Line 43: ”to other manures” is repeated twice.
8. Line 185: I would say “influent” rather than “affluent”. The same elsewhere in the text.
9. Results and discussion: it would be interesting to report a basic techno-economic assessment regarding the upscale feasibility of the tested co-digestion mixtures, especially considering the amount of substrates in the investigated area. This would increase the overall value of the manuscript. If not possible, this should be mentioned as future work to be done.
10. Conclusions are too short, the authors should recap the main findings quantitatively in this section.
11. The English language, despite being generally good, can be further refined to reach a high-quality standard.
Comments on the Quality of English Language1. The English language, despite being generally good, can be further refined to reach a high-quality standard.
Author Response
Author's Reply to the Review Report (Reviewer 2)
Dear reviewer,
All the changes made to the manuscript are highlighted in yellow throughout the text. As requested by the reviewer, the manuscript was proofread in English, and the editorial certificate is available for access. Below are the reviewer’s comments with the respective changes and comments from the authors.
Introduction:
L 58-63; L 67-69; L 79-84
Two relevant references [23,24] were inserted in the introduction (L 54).
The aim and innovation of the manuscript was revised (last paragraph of the introduction).
Materials and methods:
Figure 1. L118
The term “biofertilizer” was replaced with “digestate.”
L 138
The organic loading rate (OLR) was added to the text.
L 140-145 (solid separation)
The point was addressed with the reinforcement of literature.
Figure 2. L 191
The figure was inserted after the revision. “The portable gas analyzer employed in this study: a Gasboard-3200L”
Results and discussion:
Table 2. L 210
The term “affluent” was replaced by “influential.” The term was replaced throughout the text.
We thank the reviewer for this suggestion and inserted the VS/TS ratio. The C/N ratio of the effluent was added.
The acronym “TKN” was replaced by “NKT.”
Figures 2 and 4
Figures 2e, 4a, and 4b (original manuscript) were removed due to data repetition.
A graph of daily methane production was added (Figure 5). Total alkalinity was presented with partial and intermediate alkalinity (Figure 4).
Conclusions:
L 364-372
The conclusions were rewritten. We inserted the main results and a new paragraph with suggestions for future studies.
We would like to thank the reviewer for evaluating our manuscript. We attempted to address all the reviewer’s concerns adequately and believe that our article improved considerably. We look forward to your final response and willing to answer any further questions.